# SARS-CoV-2-Infection (COVID-19): Clinical Course, Viral Acute Respiratory Distress Syndrome (ARDS) and Cause(s) of Death

**DOI:** 10.3390/medsci10040058

**Published:** 2022-10-10

**Authors:** Giuliano Pasquale Ramadori

**Affiliations:** University Medical Clinic, University of Göttingen, Robert-Koch-Strasse 40, 37075 Göttingen, Germany; giulianoramadori@gmail.com

**Keywords:** SARS-CoV-2 infection, dehydration, hypoalbuminemia, pulmonary hypoxia, hyaline membrane, pulmonary engorgement, lung weight, acute respiratory distress syndrome, diffuse alveolar damage (DAD), heart failure, cause of death

## Abstract

SARS-CoV-2-infected symptomatic patients often suffer from high fever and loss of appetite which are responsible for the deficit of fluids and of protein intake. Many patients admitted to the emergency room are, therefore, hypovolemic and hypoproteinemic and often suffer from respiratory distress accompanied by ground glass opacities in the CT scan of the lungs. Ischemic damage in the lung capillaries is responsible for the microscopic hallmark, diffuse alveolar damage (DAD) characterized by hyaline membrane formation, fluid invasion of the alveoli, and progressive arrest of blood flow in the pulmonary vessels. The consequences are progressive congestion, increase in lung weight, and progressive hypoxia (progressive severity of ARDS). Sequestration of blood in the lungs worsens hypovolemia and ischemia in different organs. This is most probably responsible for the recruitment of inflammatory cells into the ischemic peripheral tissues, the release of acute-phase mediators, and for the persistence of elevated serum levels of positive acute-phase markers and of hypoalbuminemia. Autopsy studies have been performed mostly in patients who died in the ICU after SARS-CoV-2 infection because of progressive acute respiratory distress syndrome (ARDS). In the death certification charts, after respiratory insufficiency, hypovolemic heart failure should be mentioned as the main cause of death.

## 1. Introduction

More than two and a half years after the onset of the pandemic, more than 520 million people have contracted the SARS-CoV-2 infection and more than 6.3 million people have died [1]. However, the true number of deaths might be much higher [2,3].

While the age of the patients hospitalized in Wuhan was mostly under 65 years [4,5,6,7,8] the first children with a mean of 3 years of age were observed from 7 January to 15 January 2020 [9] with an ICU admission, nine more cases of children observed in January/February 2020 were described [10]. Children continue to be infected and eventually die also in the Western countries [11,12].

In most the non-hospitalized cases but also in hospitalized persons who died of SARS-CoV-2 infection, the cause of death has been attributed to respiratory insufficiency (ARDS) with the classical CT scan finding of the chest (without the use of intravenous contrast medium). It shows the so-called ground glass opacities (GGO), which are supposed to be due to directly induced viral damage. It is, however, important to reconstruct the chronologic sequence of clinical events leading to the changes detected by radiological imaging of the lung and to correlate them with the available autopsy findings published until recently.

Most of the autopsy reports are based on data obtained mainly in the lung of hospitalized patients whereas data collected at the onset of symptoms and during the prehospitalization time are scarce.

Treatment guidelines mostly concern the antiviral treatment and the suppression of the acute-phase reaction with experimental drugs, starting at the beginning of the hospitalization before entering and/or during the ICU stay [13,14,15]. 

Compared to the high number of therapeutic trials performed on hospitalized patients, much less attention has been paid to the treatment of initial symptoms in order to prevent physical deterioration and the need for hospitalization [16,17,18,19,20,21,22].

The first person who officially died of COVID-19 in Italy in the village of Vo’Euganeo on 20 February 2020 was a 78-year-old man [23]. Thereafter, most of the patients who died of COVID-19 in Italy were older than 80 years [24,25,26]. The same events were observed in Germany, in Denmark, Sweden, and other countries of the Western world [27,28,29,30,31] during the first wave of the SARS-CoV-2 pandemic, where more than two thirds of the people who died of COVID-19 were old or very old.

This has been explained by the special frailty of the older patients due to the presence of different comorbidities such as hypertension, diabetes, and chronic lung or kidney disease [32,33,34,35]. Most of the old persons who were infected and died lived in assisted retirement facilities [36,37,38,39,40].

Autopsy studies investigating the pathophysiology of asymptomatic/oligosymptomatic infection [41] and clinical course of the disease up to the cause of death were started late, and viral replication was supposed to be responsible for multiple organ failure.

## 2. Similarities of the Clinical Course of Influenza- and the Clinical Course of Symptomatic SARS-CoV-2 Infections

The clinical course of SARS-CoV-2 and of influenza infection is quite similar [42,43,44,45]: runny or stuffy nose, headache, dry cough, sore throat, fatigue, muscle pain, fever of various degrees and duration, inappetence, vomiting, and diarrhea which may then lead to the development of shortness of breath or difficulty of breathing. To these symptoms oliguria and loss of body weight can be added [46].

In many cases, because of the fear of infection transmission within the family members as soon as the oximeter shows a pO2 < 93%, help in the emergency room of a hospital is sought even without symptoms of shortness of breath or increase in breath frequency.

At this time, several days after the onset of symptoms, routine chest X-ray studies may, at first, result normal and CT values of PCR analysis may suggest a low viral load.

Chest CT scans performed at the same time or one or two days later may, however, show ground glass opacities in the subpleural dorsal and in basal portions of the lungs suggesting a risk of progress into acute respiratory distress syndrome (ARDS).

A further decrease in pO2 may lead to the decision of mechanical ventilation for the patient [47,48], with the diagnosis of ARDS often depending on setting and circumstances [47] which may also have been influenced, at least in part, by the administering of experimental drugs.

Although several clinical patterns have been reported, fever and dry cough are the most frequent symptoms reported at the onset of the disease (20% of the PCR-positive persons) with further worsening 6–9 days after the onset of the first symptoms characterized by dyspnea of different severity (5%) leading to hospitalization [49,50] and eventually to the ICU [50].

Anamnestic information given by the patients themselves or by their relatives concerning the onset, the clinical development, and the duration of the symptoms in the pre-hospitalization time have been often unprecise.

In fact, as many old patients were brought to the hospitals from assisted care facilities it was very difficult to ask about the grade and duration of fever, fluid, and calorie uptake, changes in body weight since symptoms first appeared, and which drugs were regularly administered. As the modern doctor has less and less time for contact with his patients and with their families [51,52], this may have been even more difficult during the first wave of the SARS-CoV-2 pandemic, because of the exceptional situation for home care by the home doctors, in the emergency room due to overcrowding, and the fear of contagion by the diffuse lack of preventive measures among the healthcare personnel.

Due to the physical conditions of the old, infected patients, who normally require a careful examination of the skin and mucosal surfaces, hydration status examination has often been impossible and therefore not reported [51,52].

In many cases, immediately after admission, patients have been sent to the radiology department for diagnostic chest CT scan [53].

This was due, at least in part, to overcrowding of the emergency rooms, because of the lack of time to wait for the virological diagnostic procedure (where available). Overcrowding also led to the further delay of administration of fluids and nutrition [54,55,56].

The findings in the CT scan of the chest often have been the main support for further clinical decisions, e.g., hospitalization and eventually mechanical ventilation.

## 3. SARS-CoV-2 Infection, Clinical Course in Hospitalized Patients, and Serum Cytokine/Chemokine Levels

The administration of antiviral drugs and monoclonal antibodies and even more frequently of corticosteroids and of anti-cytokine antibodies have been considered a priority [8,31,54,55] if administered early [56,57], especially in hospitalized SARS-CoV-2 infected patients.

Two early publications [7,58] concentrated on the changes in serum levels of inflammatory cytokines, mainly interleukin-6, TNF-alpha, interleukin-1-beta, and of the chemokine interleukin-8 during hospitalization. In the first publication, a decrease in the level of acute-phase cytokines IL-6 and TNF-alpha was observed in the serum of patients who were discharged from the hospital. The IL-1-Beta serum level was slightly elevated only in 3 of 27 patients studied [7]. In the second publication [58], the serum levels of IL-6, TNF-alpha, and IL-8 were significantly elevated in COVID-19 patients, IL-6 being the most prominent cytokine (all in the order of magnitude of picogram/mL). The Il-1-beta-serum level was always low or at the limit of detection.

Therefore, a difference in the behavior of the serum levels of these acute-phase cytokines was observed and “different cutoffs were chosen for further statistical analysis” [58].

Especially, elevated IL-6 and IL-8 initial serum levels were closely correlated with severe disease with signs of organ failure showing disastrous lung imaging, reduction of creatinine clearance, the need for vasopressors, and the use of mechanical ventilation. The authors found that the serum level of the cytokines was differently associated with comorbidities.

The first determination of elevated IL-6 and TNF-alpha serum levels after hospitalization was indicative for the prognosis of the patient, with higher levels having a poor prognostic meaning.

Under the effect of experimental drugs such as IL-6 receptor antagonists, corticosteroids, or remdesivir, there was a fluctuation of IL-6 serum levels with a potentially positive effect on prognosis. Changes in viral load under therapy were not reported.

The discussion about the source of the acute-phase cytokines and of their role and meaning within the clinical picture of COVID-19 focused more on the possible use of the measurement of serum levels to determine the prognosis.

### Possible Explanation for Increased Serum Levels of Acute-Phase Cytokines/Chemokines as a Sign of Tissue Damage during Hospitalization

The main acute-phase cytokine, Il-6, is mainly synthesized by macrophages recruited at the site of “aggression” and tissue damage. The chemokine IL-8 is also locally synthesized by every cell under “attack” and contributes to the local recruitment of inflammatory cells.

The increase in IL-8-serum levels under such acute-phase conditions, however, is more dependent on the degree of Il-6-serum level, which induces production of the chemokine in the liver [59] by directly acting on the hepatocyte. As a consequence, the IL-8 serum level, however, has to be high enough to induce increased release of granulocytes from the bone marrow [60,61].

In fact, especially in viral diseases, such as COVID-19, the serum levels of IL-6 and consequently of IL-8 do not seem to be high enough to induce the massive increase in the leukocyte count which is characteristic for bacterial infections or for other acute clinical situations [62,63,64,65].

A great amount of data concerning the prognostic significance of serum levels of different chemokines, cytokines, and other serum proteins has been generated mostly by analyzing samples from hospitalized patients [66,67,68,69,70,71,72,73,74,75,76,77,78,79,80,81,82,83,84] who, in most of the cases, received experimental drugs immediately on admission [7,47,55].

French authors [75] followed the laboratory changes observed in 162 patients during their stay in the hospital. They showed an increase in the number of patients developing acute kidney injury (AKI) from 0% on admission 21.4% at day 14. At the same time, increased osmolality and increased CRP serum levels were observed, while the albumin serum level (below 25 g/L) continuously decreased.

Those data [75] together with those of Xie [39] strongly suggested hypovolemic pre-renal insufficiency as also described by Sise et al. [80] and by Bowe et al. [81].

The replication of SARS-CoV-2 in the respiratory tract and in other organs such as the kidney has been suggested to be responsible for organ “injury”. The continuous determination of viral presence during hospitalization has, however, been rarely reported.

The work published by Xie et al. [39] shows interesting aspects which derive not only from the description of the changes observed during hospitalization in COVID-19 patients but also from the comparison with patients admitted to the hospitals because of influenza infection in the 3 years preceding the COVID-19 pandemic. Although the admission characteristics of the influenza-infected (75% white and 25% black) and of those COVID-19 patients (50% white and 50% black) were quite similar, in white patients with COVID-19, mortality, although lower than in the black population, was more than three times higher than in the influenza-infected patients. This was because of worsening of the clinical conditions occurring after hospitalization [39]. This finding may suggest that clinical worsening in SARS-CoV-2-infected, hospitalized patients may not only be due to viral “aggressiveness” but, instead, to insufficient fluid treatment.

## 4. Mortality Risk Scores in Symptomatic, Hospitalized, SARS-CoV-2-Infected Patients

Mortality risk scores have been developed, by different means, analyzing clinical data and laboratory parameters derived from blood and sometimes urine samples taken on admission to hospital, mostly under the conditions mentioned above [85,86,87,88,89,90,91,92].

Although hypoalbuminemia was repeatedly described [93] to be of crucial importance as a negative prognostic marker independent of comorbidities, the importance of sequential measurements of albumin serum levels in the emergency room and during the hospital stay has been, however, underestimated [94,95,96,97].

The increased use of diuretics, which are routinely given on the assumption that respiratory distress accompanied by ground glass opacities (chest CT scan) up to the development of acute respiratory distress syndrome (ARDS) is a kind of non-cardiac “edema” of the lung, may aggravate the general clinical status of the patient [98].

The frequent need for vasopressors, often mentioned in guidelines, may be a consequence of hypovolemia leading to pre-renal kidney “injury” [99].

Compared to patients with influenza infection, patients who were diagnosed with COVID-19 were found to have an increased risk of extrapulmonary organ failure associated with an increased use of “health resources”.

The main question is therefore whether this development after hospitalization is due to viral replication or to the increased administration of drugs [48], which seem to aggravate, in many cases, preexisting conditions such as hypoalbuminemia [86,93], dehydration [29], and consequently, tissue hypoxia and tissue damage. These are eventually the basis for hemodynamic changes leading to most of the complications with the need for routinely administered [34,48] catecholamines/vasopressors especially in older persons. Peripheral “generalized” tissue damage (including vasa vasorum [100]) is also the cause of the local and systemic inflammatory response (inmate immunity reaction [101]).

In most non-hospitalized cases but also in hospitalized persons who die of SARS-CoV-2 infection, the primary cause of death is respiratory insufficiency (ARDS) which is supposed to be due to directly induced viral damage, but pulmonary embolism is also considered to be a frequent complication in ICU patients [102]. According to the Berlin definition of ARDS [103], hypoxia without a cardiac cause represents *the conditio sine qua non* for the diagnosis of acute respiratory distress syndrome (ARDS). This is the case when hypoxemia occurs in COVID-19 patients showing diffuse ground glass opacities in the CT scan of the chest [98].

## 5. How to Define the Cause of Death

To define the cause of death, it is important to reconstruct the chronologic sequence of clinical events, of the changes detected by radiological imaging, and to correlate them with the available autopsy findings published until recently.

Most of the reports published so far are based on data obtained from patients who died after hospitalization whereas data collected at the onset of symptoms and during the pre-hospitalization time are scarce [104].

As mentioned above, the beginning of the clinical course of SARS-CoV-2 and of influenza infection is quite similar [13,16,17,19,38,40,97].

The similarity of the initial symptoms between SARS-CoV-2 and influenza infection has remained unchanged after the delta variant of the SARS-CoV-2 has been replaced by the omicron variant [105]. However, even if the symptoms seem to be less severe with fewer hospital admissions, the average number of deaths may have remained constant [106].

In many cases, because of the fear of infection transmission within family members as soon as the oximeter shows a pO2 < 93%, help in the emergency room of a hospital is sought even without symptoms of shortness of breath or increase in breath frequency.

At this time, several days after the onset of symptoms, chest X-ray studies may, at first, result normal and CT values of PCR analysis may suggest a low viral load.

A chest CT scan performed at the same time or one or two days later may show ground glass opacities in the subpleural dorsal and in basal portions of the lungs suggesting a risk of progression into ARDS.

A further decrease in pO2 may lead to the decision of mechanical ventilation [13,14] with the diagnosis of ARDS, often depending on setting and circumstances [15] which may also have been influenced, at least in part, by the administering of experimental drugs [107].

ARDS. From lung physiology to respiratory insufficiency of different origin.

## 6. Importance of Fluid Homeostasis for Pulmonal Physiology

Under normal conditions, a person of 70 kg body weight needs about 2.900 milliliters of water every day. About 700 mL of this water is necessary to preserve the alveolar lining fluid and to avoid dryness of the mucosa of the upper respiratory tract. Every day that amount of fluid is lost through exhalation.

This water originates from the alveolar capillary fluid, which the body keeps constant, indicating that the fluid volume of the alveoli is replaced about 20 times/day [108,109,110].

To understand the possible cause of the changes occurring in the alveoli before diffuse alveolar damage (DAD) with hyaline membrane deposition develops, it is important to recall the anatomical and functional peculiarities of the lung.

To exert its function, the lung needs water and oxygen [111]. Oxygen from the inhaled air is transferred to the erythrocytes through the wall of the alveolar capillaries replacing CO_2_ which will be exhaled. To this aim, alveolar tension must be reduced to 0 [108].

In the lung, venous blood from the right heart is transported by pulmonary circulation, in the pulmonary veins, to the pulmonary capillaries which build the borders of each alveolus. The second circulation is represented by the bronchial circulation originating from the aorta and from the intercostal arteries [112].

The pulmonary capillary vessels are much thinner than conventional arteries and their wall is so thin that fluid (see above) and gas can move across them.

The capillary network of the alveoli builds a “sheet” which allows to reduce flow resistance and favors gas exchange. The availability of the capillary segments may vary according to the transmural pressure difference between the inside and outside of the vessels.

When this is low many capillaries are closed. They may, however, be rapidly recruited to satisfy increased blood flow (low-pressure high-flow system). This flexibility is necessary as the lung cannot control the cardiac output. When the capacity of the heart to pump venous blood into the lung is reduced, splanchnic stagnation of venous blood occurs.

When pulmonary congestion develops, a reduction in blood volume in the systemic circulation takes place, the heart (right and left) has less blood volume to pump but changes in heart size may not be apparent in an X-ray study.

Human organs consist of up to 80% of water [113]. Water is essential for the formation of the secretions of the salivary glands, of the stomach, of the liver, and of the pancreas which allow food digestion and the passage of nutrients from the intestine into the portal blood. Water is important for the secretion of the glands of the upper respiratory tract and of the bronchial mucosa which represent the main barrier against noxious agents present in the air [114].

This must be taken into consideration when supportive measures to improve the prognosis of COVID-19 patients are being considered [27,115,116,117].

Water is, in fact, very important not only for humidification of the exhaled air but also for constant availability of the alveolar lining fluid (ALF) containing the surfactant factor in the alveolar wall. ALF is indispensable for the passage of oxygen to and of CO_2_ from the pulmonary capillaries [61,118]. The proper hydration status of the endothelial cells is extremely important to avoid shrinking of those cells in areas such as the lung capillary where the endothelial cell represents the barrier for the intravasal fluid and prevents the passage of plasma components into the alveolus [27,119,120].

Of course, water is also of vital importance for the physiological excretory function of the kidney and for the proper function of the brain.

Thirst is the central sensation which is triggered when dehydration induces 1–2% of body mass loss [116].

This induces a wide range of extracellular (decrease in blood volume and arterial pressure) and intracellular changes. 

With increasing age, dehydration can continuously develop because of the reduced sense of thirst, reduced mobility and, again, uptake of diuretics. “Chronic” hypertonicity of the blood due to dehydration has been described in older people. It is a risk factor for morbidity and mortality, and it must be kept in mind when approaching an older patient [27,113], especially if a fever above 38 degrees centigrade of several days’ duration aggravates this condition.

The geriatric population faces two main hydration-related conditions which can lead to the need for help from the doctor namely:(a)Dehydration when the total body fluid is significantly reduced compared to the age-specific hydration status (which is already reduced compared to that of the young);(b)Excess of body fluid (overhydration) when the amount of fluid in the body is higher than normal.

The stability of this equilibrium decreases with increasing age and a few hundred milliliters of fluid may switch the hydration status into one direction or the other [111,112,113,115].

Fluid loss can be caused by exposure to higher temperatures in the environment without compensation, by use of diuretics and, more importantly, by protracted high fever and inadequate fluid intake.

Oral intake of fluid for compensating fluid loss under such conditions may be inadequate and prolonged, moderate intravenous fluid administration may become quickly necessary to revert the established compensatory mechanisms to the dehydration condition before homeostasis is reestablished.

The consequence of fluid loss is the reduction in systemic blood volume, hyper concentration with appearance of symptoms such as dizziness, confusion (or delirium, brain fog), seizure, and eventually stroke and death [112,113,114,115].

Dehydration in addition to reduced blood volume and eventually hypoalbuminemia may be responsible for other ominous symptoms, namely hypoxia and dry cough [115] but can also predispose to diffuse thromboembolism, especially in the lung [102,115,116,117,121,122].

This phenomenon may be the cause of progressive “sequestration” of blood accompanied by the progressive weight gain of the lung and eventually reduction in blood volume and weight reduction in other organs, especially in the kidney [123].

The reduction in blood volume finally leads to heart failure, the definitive cause of death [124] like what happens in cases of malignant hyperthermia where fluid loss causes shock with increased lung weight [125].

## 7. First Clinical and Experimental Report of the Consequences of Dehydration for Pulmonary and Systemic Circulation

In 1941 a pivotal report was published by Harry A Davis [126] from the department of Surgery of School of Medicine of Louisiana State University and Charity Hospital of Louisiana, New Orleans, in Archives of Surgery. He reported the clinical and autoptic findings of a 42-year-old male patient who was admitted to the hospital because of intestinal obstruction and massive dehydration due to severe and frequent vomiting. The patient was clearly in a state of circulatory failure with a systolic pressure of 75 mmHg and a pulse rate of 120 per minute. The patient died four hours after admission. Besides a 28 cm length of dark red strangulated jejunum with signs of gangrene, all the other viscera except the lung were reduced in weight. The lung was heavy and moist and numerous petechial hemorrhages were present in the visceral pleura. Microscopically, the alveolar capillaries were markedly distended and filled with closely packed red blood cells.

Furthermore, the alveoli were filled with fluid and extravasation of blood was present in many areas [126].

This experience stimulated the author to perform a study on 17 dogs (6 dogs were studied as a control group) with a body weight between 5 and 10 Kg. A total of 25 cc of 25% sodium chloride/Kg body weight were injected subcutaneously into the dog’s hindleg. Two or three hours after injection the blood pressure began to progressively decrease accompanied by a progressive increase in marked edema at the site of injection until death which occurred between 5 and 10 h after injection. Outside of the edematous area of the injected muscle, the subcutaneous tissues elsewhere were dry and sticky. While the left ventricle was empty, the right ventricle was filled with thick partially coagulated blood.

While the pleural cavities were dry, the lungs were brick- and purple-red showing numerous petechial hemorrhages in the visceral pleurae. The surface of the lungs exudated dark venous blood after cutting. The alveolar capillaries were dilated with closely packed erythrocytes, many of which were found in the capillary spaces because of the rupture of the capillaries. The alveoli were filled with clear acellular fluid. The extent of the alveolar damage was correlated to the duration of the shock situation.

The animals which died quickly had less alveolar fluid (edema) than those which survived longer due to the hypotension. In a few animals some macrophages were detectable in the alveoli.

Davis concluded that dehydration is sufficient to explain the changes observed, most strikingly in the skin, the lungs, and the weight reduction of the other organs and that there was no need to suppose a role for “toxic” substances.

He supposed, instead, that hemoconcentration is accompanied by a great reduction in oxygen content in the blood.

## 8. Other Experimental Models of Pulmonary Distress and Hyaline Membrane Diseases

Many publications dealing with experimental acute lung injury (ALI), a definition used as a synonym of Acute Respiratory Disease Syndrome (ARDS) and having diffuse alveolar damage (inflammatory infiltrates, thickened alveolar septum, and deposition of hyaline membranes in the alveolar walls as the histological basis), have been published in the last decades [123,127,128,129,130].

However, no animal model was considered to reproduce all the pathologic features of human DAD which is characterized by: (a) an early exudative phase accompanied by accumulation of neutrophils in the vascular, interstitial, and alveolar spaces (“neutrophilic alveolitis”), (b) deposition of the so-called hyaline membranes due to the deposition of fibrin and other proteinaceous debris at the alveolar site, as a sign of the disruption of the alveolocapillary membrane, (c) interstitial thickening, and (d) formation of microthrombi interpreted as evidence of endothelial injury and activation of the coagulation cascade.

The lack of reproducibility of the human ALI has been justified by the fact that the human pathology may be caused by a combination of factors which cannot be reproduced in animal models [123,127,128,129,130].

Hyaline membrane formation was not in focus anymore [131,132,133,134,135,136].

All the experimental models concentrate on the functional lung damage and disregard the systemic consequences of the intraperitoneal endotoxin administration caused by the reduction in fluid and food intake resulting in blood distribution abnormalities and decreased glomerular filtration [123,129,130] or by the changes in gene expression not only in the lung but also in the organs including the liver [137]. In those experiments, the expression of smooth muscle alpha-actin was reduced not only in the heart but also in the lung 24 h after endotoxin administration in a dose-dependent manner. This must be taken into consideration whenever lung myofibroblasts are studied immunohistologically using alpha-SMA as a marker antigen [138] in damaged lungs of patients who died at different times after SARS-CoV-2 infection.

Additionally, the decrease in CD31 (PECAM-1) gene expression in the inflamed areas is most probably due to the local production of TNF-alpha, which downregulates PECAM-1 gene expression in endothelial cells [139].

However, histologically, hyaline membrane formation, the hallmark of the acute respiratory distress syndrome (ARDS), was already described by Berfenstam et al. [126,132] in the lung of rabbits intoxicated with O_2_ (Figure 1) and in the placenta and in the lung of O_2_-intoxicated guinea pigs and rabbits [131].

## 9. First Viral Acute Respiratory Distress Syndrome as Post-Influenza “Pneumonia”

There are similarities of the macroscopic and clinical changes described by Davis because of dehydration to those histological findings observed in the lungs of guinea pigs and rabbits experimentally induced by O_2_-intoxication and those described by Petersdorf et al. [42], 1959, in the lungs of patients who died of “post-influenza pneumonia”.

While the number of emergency room visits for non-respiratory illness remained constant, the number of respiratory illnesses increased sharply with a peak during the last two weeks in October when 170 and 157 patients were seen [42]. It was the year 1957 when the Asian influenza epidemic arrived in New Haven in late September. Most of the infected persons had mild disease but 91 of those patients presenting pulmonary infiltrates, who were seen at the emergency facility of the New Haven Hospital, were hospitalized during the six-week period from October 1957. This was a six-fold increase in the hospitalization number compared to that of the previous year.

Among the 53 males, 26 were older than 50 whereas 24 of 38 females were between the ages of 16 and 40. The authors commented on these numbers, saying that old men were prone to developing pneumonia following influenza while females were most vulnerable during the child-bearing period.

In total, 40 of 91 hospitalized patients were black, while only 15% of the medical ward patients were black. Bacteriological investigation demonstrated bacterial infection in 38 patients while the other 43 patients with radiological or autoptic signs of “pulmonary infiltrates” were diagnosed with pneumonia of undetermined etiology. In total, 10 patients were diagnosed as having acute tracheobronchitis.

Virological diagnostics consisted of investigating blood samples for the hemagglutination inhibition test and throat lavage fluid for virus culture.

A fourfold increase in the hemagglutinin-inhibition test was found in the blood of most of the patients who were tested, independently of the supposed cause of the pneumonia suggesting that the Asian influenza virus was a causal factor in most of the patients.

Symptoms preceding the visit to the emergency room were quite similar whereby 90% of the patients had a fever from 38 to 39 degrees Celsius.

In 19 patients the body temperature was even higher at presentation and persisted 48–72 h after admission.

Patients with comorbidities such as heart (including three patients with hypertension) or chronic lung diseases, diabetes, and alcoholism were found to be more susceptible to post-influenza pneumonia as was the case for 10 pregnant women. Only 36 of the 91 patients were free of comorbidities or pregnancy. In total, 11 of the 91 patients died. Four of them (68, 72, 76, and 77 years old) were suffering from chronic diseases and pneumonia played an important causal role in their death. The other seven patients who died were young healthy adults who succumbed to an infection characterized by: acute onset, high fever, severe dyspnea, leukopenia, bloody sputum, anoxia, and circulatory failure. Autopsy findings in a 19-year-old student who died 48 h after admission were identified [140] as pathognomonic. The macroscopy of the lungs showed edematous red organs with a weight two times the normal weight. Microscopy showed severe hemorrhagic pneumonia with necrosis and hyaline membrane formation.

All patients showed a similar clinical pattern with mild upper respiratory disturbances, systemic symptoms lasting for several days followed by fever, cough, dyspnea, bloody sputum, and leukopenia. The authors underscored the fact that the pathologic picture consisting of acute tracheobronchitis, diffuse involvement of the pulmonary parenchyma, necrosis of the alveoli, and hyaline membrane formation with acellular hemorrhagic exudate was quite uniform not only in the New Haven patients but also in those who died of the same disease in the USA and England.

This report suggests many similarities with the actual pandemic situation and describes similar autoptic findings with the central histologic hallmark, namely the presence of the hyaline membranes, which were known to represent fibrin deposition [132,133] and has been reported not only in premature children but also in human adults [133,134,135,136]. As it was described by Fujikura [132], fibrin deposition can be observed not only in the lungs of newborns but also in normal-term placenta [135]. Perivillous fibrin deposition in the placenta of COVID-19-positive pregnant women after delivery has been recently reported together with trophoblast necrosis and histiocytic intervillositis. The triad has been called SARS-CoV-2-placentitis [141].

## 10. Respiratory Distress Syndrome, Pulmonary Hyaline Membrane Disease, Becomes Acute Respiratory Distress Syndrome (ARDS)

Descriptions of ARDS cases by Ashbaugh and colleagues [142] and later in several other reports of ALI/ARDS cases [143,144,145,146,147,148,149,150,151,152,153,154,155,156,157] are strikingly similar to those clinical and autoptic descriptions by Petersdorf [42].

Radiologists attempted to correlate the severity of lung edema with patient outcomes [146,157,158].

In an article dedicated to clinical and radiologic features of pulmonary edema Gluecker and coworkers thoroughly describe the clinical characteristics of the “PERMEABILITY EDEMA WITH DAD” and clearly describe the corresponding radiological pattern as determined by conventional X-ray or CT scan of the chest [159].

It is clear from the beginning that this part deals with the clinical picture of ARDS, a term used for acute or subacute pulmonary “lesions” accompanied with hypoxemia.

It must be made clear that: (a) the lesions observed during radiography are not caused or influenced by heart failure and (b) ARDS occurs without an increase in pulmonary capillary pressure.

The authors divided ARDS in two major groups depending on the possible different etiologic mechanisms which may lead to their development:ARDS due to underlying (assumed) pulmonary disease.ARDS secondary to extrapulmonary disease which manifests with interstitial edema and alveolar collapse.

The authors justify the differentiation with the different implications for the treatment of patients, for example, in the cases of sepsis, acute pancreatitis, severe trauma, or blood transfusions compared to a more directly induced damage of the alveolar and vascular endothelium of the lung resulting from the exposure of the cells to chemical agents, infectious pathogens (such as bacteria or viruses), gastric fluid, or toxic gas supposed to destroy or severely damage the tissue.

ARDS may undergo three stages:

(a) The first stage is characterized by interstitial “edema” (exudative) with a high protein content that rapidly fills the alveoli and is associated with hemorrhage and formation of hyaline membranes.

The words exudative, edema, or pneumonia are not radiologic findings but correspond with a deduction, which is not always possible at the time the radiologic investigation is performed. The radiologist can only speak of ground glass opacity (GGO) without an apparent cardiologic cause.

Furthermore, there are early cases of ARDS without DAD (characterized by an “inflammatory” infiltrate) which should not be called pneumonia.

One important differential marker between GGO of cardiac and non-cardiac origin is the absence of Kerley lines characteristic of the increase in interstitial fluid of cardiac origin.

(b) The second stage (proliferative) is characterized by the organization of the alveolar fluid (called the exudate).

(c) The third stage is characterized by the formation of fibrotic septa.

At the CT scan, ARDS has a more peripheral and cortical distribution which may change by changing the position of the patient suggesting that atelectasis also plays a role in the inhomogeneous (regional) appearance, which is more common in ARDS than in classical bacterial pneumonia.

It is interesting to learn that recurrent “exudative” episodes can occur in the proliferative and fibrotic stages of ARDS resulting in a mixed radiologic picture.

## 11. Modern Developments on Morphological Diagnosis of ARDS as a Pulmonary Distress Status from a Different Origin: The Combination of New Imaging Techniques and Pathology

In the last 20 years, many efforts have been directed towards using a CT scan of the chest as a further prognostic characteristic. Recently, a Radiographic Assessment of Lung Edema (RALE) score has been introduced [160] to assess the extent and density of pulmonary opacities in patients with different severity grades of ARDS [161]. Additionally, by using this method there was a positive correlation with more conservative fluid therapy, a lower RALE score, and mortality.

The APACHE III score [143] has been used to differentiate two groups of critically ill hospitalized patients participating in the different therapeutic forms, conservative versus liberal, based on clinical characteristics.

It must be pointed out, however, that the clinical value of two very important parameters, namely albumin and creatinine serum level, were quite underestimated as only the albumin serum level of 2.4 gr/dL scored 6 points and creatinine of 2.2 mg/dL 7 points of 107 total points while an albumin serum level of 2.8 gr/dL scored 0 points and a creatinine serum level of 1.8 mg/dL scored 4 of 45 total points. On this basis, it is difficult to estimate the effect of different fluid administration policies in patients who were evaluated because of an increased amount of fluid in the lung (called edema) without measuring the blood volume.

Recently two-dimensional (2D) and three-dimensional (3D) reconstructions of pulmonary vasculature by using a CT scan of the chest have been used to study changes in the vascular morphology [162] or the loss of vessels in smokers who had undergone lung resection because of lung cancer [163]. The combination of histological and imaging data allowed the 3D reconstruction of the pulmonary vasculature down to a vascular radius of 0.5 mm and to establish a correlation between changes in the vasculature as assessed by histology and as measured using a CT scan.

Eckermann et al. [164] analyzed six lung samples from a patient who died of respiratory failure due to COVID-19.

A three-dimensional representation of the classical findings of DAD was generated with impressive pictures of hyaline membranes adjacent to the epithelial lining. Moderate lymphocytic infiltration and singular thrombi in small pulmonary veins were described.

Additionally, samples with high amounts of swollen, inflamed blood vessels and thick hyaline membranes were found. In some slices, capillaries filled with erythrocytes were shown.

Although in the manuscript the macroscopy of the lungs was not mentioned, all patients had heavy bluish, firm lungs with impressive congestion of the capillaries filled with packed erythrocytes [164]. All patients died at different times after SARS-CoV-2 infection. The presence/proliferation of viral particles, however, was not described.

It is in fact important to realize that SARS-CoV-2 infection is the trigger of early pathophysiologic changes. The consequences of those changes may not be attributable to the direct interaction of the virus with the lung and even less with other organs [165,166].

While there was no difference in the CT characteristics upon hospital admission between the group with the moderate type and the severe or critical type of SARS-CoV-2-infected patients (88% had GGO), a difference could be observed in the second CT scan of the severe/critical group which showed more frequently a lower (50% vs. 82%) occurrence rate of ground glass opacity (GGO), but a higher occurrence of crazy-paving pattern (75% vs. 39%) and of consolidation (81%). The occurrence rate of pleural thickening or adhesion also peaked (100%) at the second CT scan [167]. Similar results were reported for the groups of the moderate or severe/critical patients at repeated CT scans [168] by Huang Y et al.

## 12. How SARS-CoV-2 Infection of the Nose Leads to ARDS, Cardiac Arrest, and Death

Autopsies at the initial stage of the disease including analysis of the upper respiratory tract are seldom performed [169].

The autoptic findings did not show damage at the level of the organs other than the lungs [170,171,172].

The gross morphology as shown in Figure 1 (panel A) and histology (panel B) of the lung regularly found in hospitalized patients who died after SARS-CoV-2 infection, was very much like that observed in the victims of SARS-CoV-1-infection [173] as shown in Figure 2. The macroscopic picture of the congested lung and microscopic picture of DAD with the classical hallmark of the presence of hyaline membranes was also very similar.

As acute kidney injury is a clinically ominous prognostic sign in hospitalized SARS-CoV-2 infected patients, it is of relevance to note that tubular necrosis as a “marker” of hypovolemic shock has been often found [170,171,172] in the absence of viral replication.

In fact, viral infection has not been found in the kidneys although positive PCR findings were described and the presence of viral particles was suggested [170] when the electron microscopy technique was used, which, however, requires a lot of experience for plausibility checks [171,172].

Ischemic signs were not only found in the kidney but also in the heart and in the brain.

Multifocal vascular injury but no viral particles were described in the brain of 13 SARS-CoV-2 positive patients who died (11 of them died suddenly and unexpectedly) [174]. Microinflammatory changes have been also described in the retina [175].

In situ hybridization using SARS-CoV-2-specific RNA gave negative results.

Gelsomina Mansueto stated in a short communication [176]: “The lack of autopsy findings did not allow us to evaluate with greater serenity what was happening and to understand that the specific pulmonary histological pictures and the secondary multiorgan damage in the patients with comorbidities were very similar to those already observed for other viral infections”. She also underlines in her conclusion the importance of unspecific supportive measures to prevent complications which are not different from those caused by other viruses and can lead to multiorgan failure and death [176].

Several metanalyses have been recently published about the autopsy reports of the last two years and all agree that the main changes are found in the lung and that necrosis of the kidney tubules is quite often described [177,178,179]. The question of the cause of death after infection of the nasal mucosa by the SARS-CoV-2 virus [180,181,182,183,184,185,186] has been clearly addressed by Hooper JD et al. [187]. For death certification the cause of death could be divided into:Primary.Contributing.

Acute respiratory disease was most often cited as the primary cause as “COVID-19 pneumonia” or “acute lung injury” in 75% of the cases.

In 25% of the cases where COVID-19 was not the primary cause of death, COVID-19 was listed as a contributing cause in six cases.

Although 54% of the 135 patients who would later die acquired acute kidney disfunction and 35% acquired myocardial dysfunction, these organ disturbances were not reported as the primary cause of death but mostly reported as a contributing cause of death.

There was, however, no explanation for the “fluid imbalances” which likely contributed to the “congestion” and to the 3–4- fold increase in the lung weight which was frequently reported on the one side and the acquired kidney injury with hallmark of the tubular necrosis on the other.

None of the autopsy studies mentioned the amount of fluid found in the different organs and most importantly in the heart chambers and in the urinary bladder.

By measuring sublingual microcirculation for vessels smaller than 20 μm in 12 severely ill COVID-19 patients, Damiani et al. [188] found an inverse correlation between the plasma level of D-Dimer, the perfused vessel density, and the pO2/FiO2 value [189]. In that study, the D-Dimer plasma level ranged between 717 and 5536 ng/mL.

An inverse correlation between the plasma level of the D-Dimer and microcirculatory changes was found also in the retinal vessels of severe COVID-19-infected patients [190,191]. This indicates once more that the changes observed in the alveolar capillaries and those observed in other territories [191] may be attributable to the same noxae, namely dehydration conditioning tissue oxygenation followed by the emergency induction of the perivasal clotting process.

The “conservative” fluid administration [192] also influences the nutritional part of the care of the hospitalized and especially of ventilated COVID-19 patients. In fact, in the latter case, also pediatric intensive care physicians [193] are reluctant to administer calories and amino acid solutions intravenously because of the possible volume overload by an insufficient oral (naso-gastric tube) administration of calories. On the other hand, vasopressors used to sustain systemic circulation and diuresis are administered regularly in severely ill patients [34].

The vital prerequisite for administering fluid is, however, a normal albumin serum level (3.5–5 gr/dL). In fact, a normal albumin serum level is essential to assure normal blood volume, to guarantee normal erythrocyte circulation, especially in the alveolar capillaries, and to avoid extravasation of fluid especially in the lung.

This aspect of the clinical picture in severely ill patients in the ICU must be considered when therapeutic options are discussed in ARDS sub-phenotypes [34,194]. In the hyperinflammatory group, in fact, systolic pressure was under 90 mm Hg and the use of vasopressors was frequent. The discussion about liberal or conservative fluid support must be conducted based on a normal albumin serum level [192,193].

Furthermore, conservative fluid administration is often associated with furosemide administration [34], to assure sufficient urine production. This strategy, however, leads to hypovolemia and hypotension which brings into play vasopressors in the attempt to reduce hypotension. Vasopressors, however, can even worsen the ischemic consequences in the tissues and be responsible for tissue damage, the local release of acute-phase cytokines (e.g., IL-6) by the recruited macrophages, and for the local and hepatic production of chemokines (e.g., IL-8) [59] and other acute-phase proteins.

## 13. Conclusions

The cause of death in SARS-CoV-2-infected symptomatic hospitalized patients may be mainly heart failure as consequence of dehydration accompanied by lung congestion, increase in lung weight, and hypovolemia. While first local and systemic symptoms are triggered by the virus, the persistence of the SARS-CoV-2 in the upper respiratory tract is not always essential, especially for deaths after long-term hospitalization. Respiratory distress on the basis of increased lung weight is also a common finding of SARS-CoV-1 and in influenza infections.

## Figures and Tables

**Figure 1 medsci-10-00058-f001:**
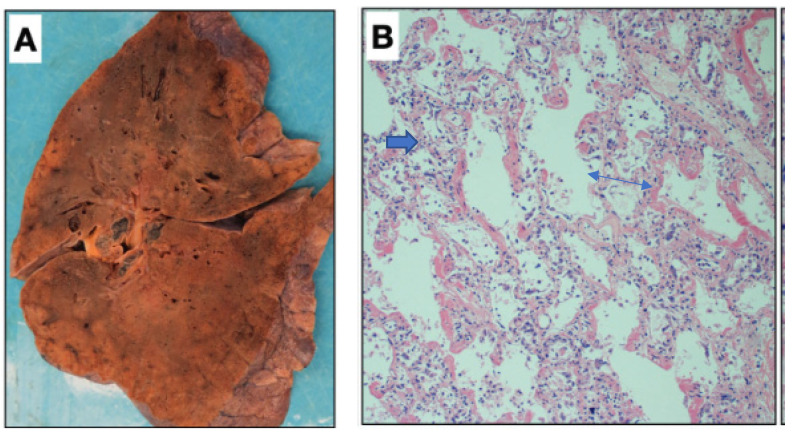
Macroscopic picture of a lung taken during a post-mortem autopsy of a patient who was admitted at the Mount Sinai Hospital and died because of SARS-CoV-2 infection. The gross appearance resembles that of a solid organ with “patchy” areas of consolidation (**A**). The microscopic picture (**B**) shows the “congestion” of the alveolar capillaries, which may represent the initial episode of the lung changes accompanied by fibrin deposits (blue arrow) of the article published by Bryce C et al. in *Modern Pathology* 2021; 34:1456–1467 [40]).

**Figure 2 medsci-10-00058-f002:**
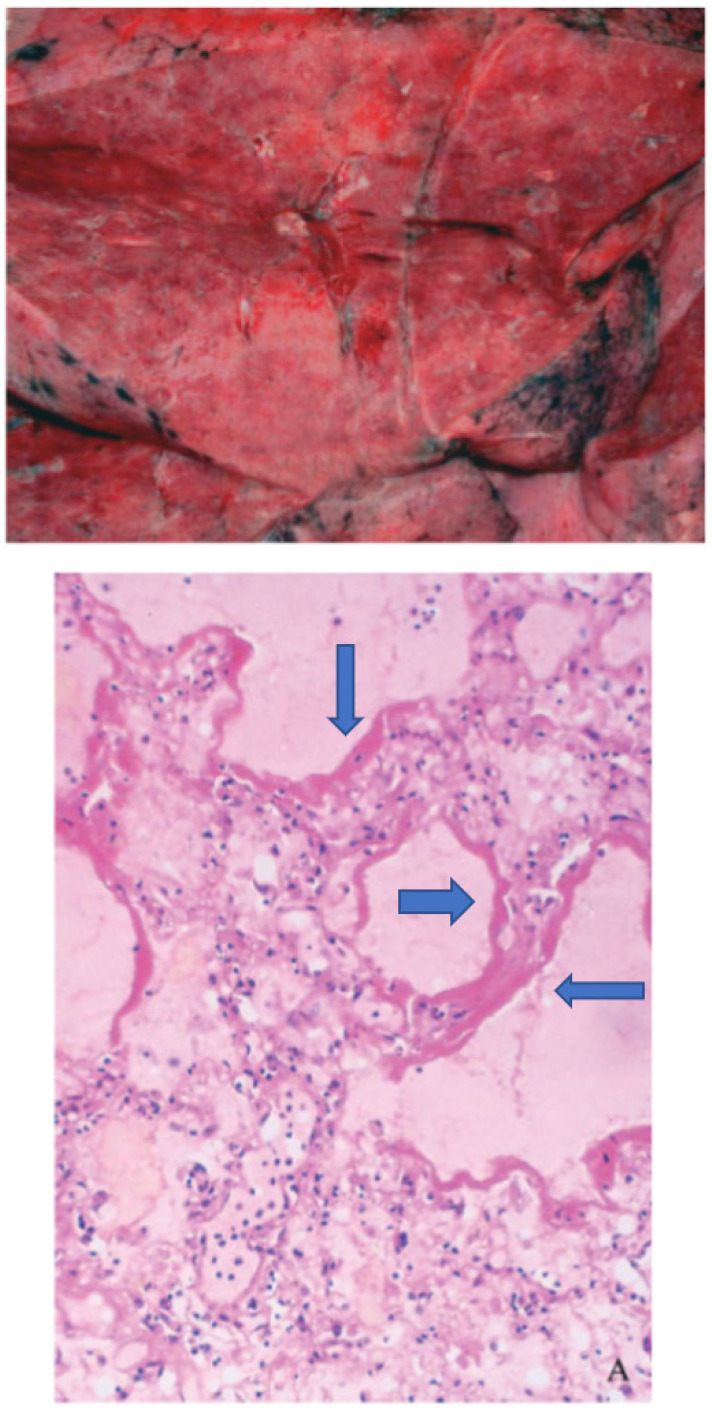
Gross morphology (upper panel) from the lung of a patient who died after a SARS-CoV-1- infection shows a congested lung while the histology picture (lower panel) shows the hyaline membranes (blue arrows). Modified from Ding et al. [173].

## Data Availability

Not applicable.

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
