# Peer review of "SARS-CoV-2-Infection (COVID-19): Clinical Course, Viral Acute Respiratory Distress Syndrome (ARDS) and Cause(s) of Death"

_medsci, 2022, doi:10.3390/medsci10040058_

Round 1
Reviewer 1 Report (New Reviewer)
Thanks to review the manuscript entitled SARS-CoV-2-Infection (COVID-19): clinical course and cause(s) of death from G. Ramadori. I read the work with many interests. The author addresses the clinical course of COVID-19 patient, as well as in relation to other viral infections, defined pathophysiology reasons of the organ impairment of COVID-19 patients and offer causes of death after COVID-19 infection. Surely, the ductus of the manuscript highlights especially preclinical conditions of patient before hospitalization is required, followed by organ impairment and therapeutic options depending on disease severity and cause of death. But the pathophysiology of COVID-19, the associated organ impairment, and disease progression are more complex than the preclinical conditions of dehydration and malnutrition in the clinical course of COVID-19 patient. These aspects are relevant points and surely underrepresented in the present discussion. And I agree, it has to be more in focus of clinical interpretation of infected patients. So, in this context some points have to be addressed.
Major points:
First, the provided abstract of the online review report differ from the abstract in the downloadable manuscript. Because of that in the manuscript the abstract is misleading because of incongruent content and missing ductus.
The ductus of the manuscript is sometimes misleading for the reader. I would prefer to rearrange it or to insert subheadings. So, it would be easier for the reader to understand the main aspects of the content. And a conclusion section would be helpful for the readership.
The clinical course due to a multifactorial immunological disease associated immune response caused by COVID-19 was leading to the therapeutic strategies of COVID-19 Patient. The preclinical conditions might by one reason for the worst outcome, especially in elder patients. But, beside to the preclinical conditions, it has to consider discussing the context more in detail to these multifactorial phenomenon’s through COVID-19 infection, e.g., adaptive/innate immune response, complement activation, coagulation, and associated disorders.
The content of section 2 and 4 are nearly congruent and only differ in provided references.
Surely, the interpretation of the causes of death is more difficult than described in the last chapter. Especially the last two sentences in the manuscript should be critical reviewed by the author.
Please, rearrange and label the tables. As stated now, it is difficult to understand what you mean.
Minor points:
Check the numbering of the manuscript. There are to headings numbered with same number.
Please check text style and English spelling.
Author Response
Point-by-Point answers to the reviewer 1
First of all I wish to thank the reviewer for the invested time and for
the very constructive suggestions.
Major points:
1.the abstract has now been corrected.Compared to that published
in the preprint-version the actual abstract had to be reduced to 250
words.
2.thank you for the very helpful suggestion.Some reshaping has now
been done and a conclusion is now added at the end of the
manuscript. Because of the lenght of the actual version some of the
immunological aspects could not be further stressed.
3.I aggree that sections 2 and 4 may be somhow redundant.I hope
that the changes introduced may have reduced this impression.
4.Thank you for the kind comment about the „provocative“
sentences. They should help to take this aspect of the causes of in-
hospital worsening of the clinical conditions more seriously.
Possibly, they should also helpto go through the treatment
guidelines for hospitalized patients.The conclusions are based on the
interpretation of the data of the literature and of the authors` own
clinical and scientific experience.
5.Thank you very much again for the comment about the table.The
heading of the table has now been changed and I hope that it is now
clear that both parts deal with physiology (I) and the pathological
consequences of quality changes of ALF.The latter may contribute
(especially definciency of water) to understanding of the early
changes taking place in the alveoli because of dehydration.
Minor points:
1.thank you so much for the suggestions.I hope that the changes of
numbering and the new headings are now satisfactory.
Reviewer 2 Report (New Reviewer)
I read with interest the article entitled " SARS-CoV-2-Infection (COVID-19): Clinical Course and Cause(s) of Death”.
The article is aimed at clarifying the pathophysiological mechanism that first leads to the development of the disease and then to death from COVID-19.
Overall the article is well written but some changes are necessary to make it worthy for publication.
Major concerns
a) The article is presented as a review but the methods of selection of references are not described. If it is not possible to report this data, it is suggested to speak of "narrative review".
b) It should be clarified that unfortunately only after a few months from the start of the pandemic it was possible to carry out autopsies and that therefore some elements characterizing the pathology were understood late. [Nioi, Matteo, et al. "Autopsies and asymptomatic patients during the COVID-19 pandemic: balancing risk and reward." Frontiers in Public Health 8 (2020): 595405.]
c) I believe it is necessary to widen the vascular alterations also of a histological. [Faa, G., et al. "Aortic vulnerability to COVID-19: is the microvasculature of vasa vasorum a key factor? A case report and a review of the literature." Eur Rev Med Pharmacol Sci 25.20 (2021): 6439-6442.].
d) The article is very long and in its context there are several comparisons with pathologies in which there are alterations similar to those found in COVID-19. The author should either shorten this part significantly or acknowledge in the title of the paper that the article also presents such comparisons.
e) More emphasis should be placed on new information or new hypotheses emerging from the author's analysis.
Author Response
Point-by-Point answers to the reviewer II
I wish to thank the reviewer for the time invested in reading the
manuscript and for the very constructive suggestions.
a)the criticism is justified.I do admit that the selection of the of the
publications is mainly based on accessibility,my own clinical and
scientific experience after more than 45 years of clinical work as
internist, gastroenterologist, and chief of an infectious disease ward.
I hope that some judgements based on own knowledge (without
literature citation) are acceptable especially if they are supported by
publications.
b)I aggree with the suggestion and the opinion paper has now been
introduced into the manuscript.
- c) the suggestion is vey important.The article is now mentioned.
d)the point is well made.Thank you again.I introduced some changes
in the title „viral ARDS“. I did not mention Influenza and SARS-CoV-1
because I thought that the title would be too long.
- e) a conclusion has now been introduced at the end of the article.It
should be clear that lung congestion with increase of lung weight ist
the main autopsy finding and the characteristic change of
dehydration.The second is tubulus necrosis in the „ischemic“ kidney
(not mentioned).
1) Thank you again.The abstract has been reshaped and the changes
suggested have been made.
2) I hope that the number of mistakes is now significantly reduced.
3)the reviewer is right.I hope that the changes are now satisfactory.
4) I hope that the journal will help to introduce the numbering.
5)I hope that all the „autoptic“ have now been replaced by „autopsy“
6) for all the figures a permission was given
7) see answer to a)
8) page 4 has been rephrased,hopefully with success.
9)the reviewer is right.I hope that the corrections have solved that
problem
10) the heading of the table has now been changed and that the two
parts should be now more informative than they were before.
11.the reviewer is right .Some references have now been introduced
into the text
12) Thank you fort he kind suggestion. A short conclusion is now at
the end of the manuscript
Reviewer 3 Report (New Reviewer)
1) In abstract "autoptic studies", there is the not the correct term, autopsy is the correct term. Also there are other mistakes in the syntax of the abstract.
2) All the manuscript need to improve the syntax and there is a need of an extensive English editing.
3) There are several typos or editing mistakes, like in the beginning of last paragraph of the page 1 and in the end of this paragraph in the page 2. There are word underlined.
4) The lines of the manuscript are not numbered.
5) The author repeat autoptic therm, the correct term is autopsy.
6) The author have permission to publish the CT-scan image from figure 1?. Also the same for figure 2 to 6.
7) There is not clear is the manuscript is a review or there is an original article. If the manuscript is a review, the author is mixing apparently own information with review information.
8) Information in page 4 is confuse, there is not a correct organization of the ideas and the main message.
9) There are several paragraph composed only by one sentence. The way of writing the article leaves a lot to be desired.
10) Table??? there is not a table, just disorganizing words that do not look like a table.
11) There are several paragraphs with out a reference, therefore, we can not differentiate between data from other authors, author own data or a comment from the author.
12) There is not conclusion or discussion.
Author Response
Point-by-point answer to reviewer III
Thank you very much for the time invested in this review and for the very constructive suggestions
1) Thank you again.The abstract has been reshaped and the changes suggested have been made.
2) I hope that the number of mistakes is now significantly reduced.
3)the reviewer is right.I hope that the changes are now satisfactory.
4) I hope that the journal will help to introduce the numbering.
5)I hope that all the „autoptic“ have now been replaced by „autopsy“
6) for all the figures a permission was given
7) the criticism is justified.I do admit that the selection of the of the
publications is mainly based on accessibility,my own clinical and
scientific experience after more than 45 years of clinical work as
internist, gastroenterologist, and chief of an infectious disease ward.
My method was also based on the the presence of description of macroscopy of the lung and lung weight ,in those reports where only the lung was studied.Important was also to have other organs studied and possibly if their weight was given.
I hope that some judgements based on own knowledge (without
literature citation) are acceptable especially if they are supported by
publications.
8) page 4 has been rephrased,hopefully with success.
9)the reviewer is right.I hope that the corrections have solved that problem
10) the heading of the table has now been changed and that the two parts should be now more informative than they were before.
11.the reviewer is right .Some references have now been introduced into the text
12) Thank you fort he kind suggestion. A short conclusion is now at the end of the manuscript
Round 2
Reviewer 1 Report (New Reviewer)
That you for the revised version of the manuscript. Several points which were addressed in the revision now provide a better understanding of the authors suggetsions for the reader.
But there are always now points, which must be addressed.
I think, the table is a problem in the way it is presented. It is difficult to understand what is the content or the meaning of it.
The conclusion in may opinion is not a conculsion. It offer the subjective authors opinion, and a review should provide an objective point of view of actually literature.
Author Response
Manuscript ID:medsci-1773572
Review
Title:SARS-CoV-2(COVID-19):Clinical course,viral Acute Respiratory Distress Syndrome (ARDS) and Cause (s) of Death
Authors:Giuliano Ramadori
Point-by-point answer to reviewer 1.
Dear reviewer thank you very much for your very helpful suggestions:
1.the table has now been deleted
2.the statments of the conclusion has now been weakened by the introduction of „may mainly be due“ in the first and second sentence.
Reviewer 2 Report (New Reviewer)
The article has now greatly improved. Of note, I ask the authors to replace the term "cardiac arrest" with "heart failure". According to forensic doctrine, cardiac arrest is the final stage common to all deaths and cannot be indicated in the causal analysis. I believe this point is fundamental and I apologize to the author for the failure to report it in the first round. For the rest I am satisfied with the changes made.
Author Response
Manuscript ID:medsci-1773572
Review
Title:SARS-CoV-2(COVID-19):Clinical course,viral Acute Respiratory Distress Syndrome (ARDS) and Cause (s) of Death
Authors:Giuliano Ramadori
Point-by-point answer to reviewer II
Dear reviewer,
thank you very much for the very helpful suggestions.
The definition cardiac arrest has now been replaced by „ „heart failure“ as suggested.
Reviewer 3 Report (New Reviewer)
The manuscript is a review article, but it looks like if the author mix published literature with data, images and studies from patients. Is not clear where the patients data was collected. The author did not present the Ethical Committee approval to show CT image (figure 1 and 4) (this is a major Ethical concern).
Also the tables are not in a format of table, ande there are not clear.
If the authors want to publish the review, it must change the CT image or get approval of the hospital Ethical Committee and fix table 1.
Author Response
Manuscript ID:medsci-1773572
Review
Title:SARS-CoV-2(COVID-19):Clinical course,viral Acute Respiratory Distress Syndrome (ARDS) and Cause (s) of Death
Authors:Giuliano Ramadori
Point-by-point answer to reviewer III
Dear reviewer
Thank you very much for your constructive criticisms and valuable suggestions:
1.figure 1 and 4 have been deleted
2.the table has been deleted
Round 3
Reviewer 1 Report (New Reviewer)
Thank you for the revised version. Due to a magnitude of changes during systemic infections caused e.g. by bacterial, viral, or fungal sources, it is to placative to conduct the causes of death only to hypovolamia, increase lung weight following heart failure or cardiac arrest. In the end cause of death will be heart failure or cardiac arrest independent of a life limiting diseases. The manuscript in the present way must be improve.
Author Response
Thank you very much for the constructive suggestions.
It is however clear that cardiac failure followes pulmonary congestion,which is initiated by ischemic changes in the pulmonary capillaries.
This is not due to local viral replication.This is also supported by a recent publication in Lancet Infectious Disease by Remuzzi et al.
The language mistakes have been now corrected
Reviewer 3 Report (New Reviewer)
It can be published
Author Response
Thank you very much for the constructive suggestions
This manuscript is a resubmission of an earlier submission. The following is a list of the peer review reports and author responses from that submission.
Round 1
Reviewer 1 Report
The author submitted a review article named "SARS-CoV-2 Infection (COVID-19): Clinical Course and CAUSE (S) OF DEATH" describing a variety of topics from "Acute Respiratory Distress Syndrome (ARDS):definition and possible causes" to "The Future of symptomatic HuCoV-2-infection is in Home Treatment for Hospitalisation Prevention".
In my opinion, there are a couple of problems with the manuscript the author needs to address:
- The manuscript is difficult to read, too long and should be (and could be) shortened to approx. 50%.
I would further propose to concentrate the main part of the manuscript on topics relevant to the title of the manuscript. The author should elaborate on which patients under which conditions were hospitalized, admitted to the ICU, put on mechanical ventilation and ECMO and needed further organ support like dialysis.
As correctly stated by the author, most patients with SARS-CoV-2 associated pneumonia die while on mechanical ventilation which might be a main contributor to the high mortality rate in these patients. As shown previously patients with pulmonary failure in need of mechanical ventilation usually do not die because of associated hypoxia but due to secondary complications and extrapulmonary organ failure. This topic is especially important in SARS-CoV-2 associated pneumonia and should be thoroughly discussed in a manuscript with this title.
Furthermore, the manuscript should elaborate on the relevance of pulmonary co-infections.
2) The author mentions multiple times the postmortem findings of pulmonary venous congestion from ARDS patients discussing hypovolemia as potential risk factor for this finding. Numerous other explanations like pulmonary embolism, acute cor pulmonale due to hypoxic pulmonary vasoconstriction and invasive mechanical ventilation come to mind. The author should elaborate on these as well.
3) The manuscript needs rigorous editing. Please correct the multiple typing errors and the repetitions in the text (i.e. compare page 2 line 67-91 and page 9 line 257 - 275).
Reviewer 2 Report
The review by Giuliano Ramadori entitled, “SARS-CoV-2-Infection (Covid-19): Clinical Course and CAUSE (S) OF DEATH”, presents the thoughts of the author on the clinical progression of the disease. Although it appears to represent a fair amount of work, it is very unrefined, poorly written and is a disjointed and rambling discussion of disparate disease parameters without focusing on the intent and topics indicated in the title. The review skirts and does not relate the issues of virology and immunopathology to the pathogenesis and disease development of COVID-19 but extensively expands on individual aspects of the disease presentation with minimal relation back to COVID-19. The author includes figures taken from other publications and comments with poor discussion that might be appropriate in a textbook but not in a review without permissions and clear attributions. Spelling, grammar and presentation are problematic. It is much too long. Overall, the manuscript does not answer the question raised in the title but wanders around many other subjects and as such, this manuscript is inappropriate for publication in this journal.